# SLAC and SNAC Wrist: The Top Five Things That General Radiologists Need to Know

Eliza Kompoliti [1], Mikaella Prodromou [1] and Apostolos H. Karantanas [1,2,3,*]

1 Department of Medical Imaging, University Hospital, Voutes, 71110 Heraklion, Greece; eliza_kompoliti@yahoo.gr (E.K.); prodromoumikaella@gmail.com (M.P.)
2 Department of Radiology, Medical School University of Crete, Voutes, 70013 Heraklion, Greece
3 Advanced Hybrid Imaging Systems, Institute of Computer Science, Foundation for Research and Technology (FORTH), 70013 Heraklion, Greece
* Correspondence: akarantanas@gmail.com or karantanas@uoc.gr

**Abstract:** Wrist osteoarthritis (OA) is a common painful condition that affects the patient's quality of life by limiting the range of wrist motion and grip strength. Wrist OA often results from scapholunate advanced collapse (SLAC) or scaphoid nonunion advanced collapse (SNAC). Early diagnosis of SLAC and SNAC is crucial because it affects treatment planning. Thus, radiologists should be able to interpret the early imaging findings. This pictorial review discusses the pathophysiology and the clinical symptoms of SLAC and SNAC and presents the imaging findings with emphasis on the proper imaging algorithm. Finally, it focuses on the treatment according to the degenerative status of each of these patterns.

**Keywords:** SNAC wrist; SLAC wrist; wrist osteoarthritis; CT-diagnosis; MR imaging/diagnosis; arthrography

## 1. Introduction

The wrist is perhaps the most complex joint in the body. Wrist osteoarthritis (OA) occurs in about 95% as a periscaphoid problem, and it can result in severe functional disability [1]. Scapholunate advanced collapse (SLAC) and scaphoid non-union advanced collapse (SNAC) are the most common patterns (55%) of wrist OA, which follow a predictable sequence of degenerative changes [1–4]. Imaging plays a crucial role in the diagnosis and monitoring of SLAC and SNAC wrists. Therefore, understanding the pathogenesis and recognizing the specific imaging findings are essential for the general radiologist. In this review article we focus on imaging findings in an effort to expand radiologists' awareness of the need for early diagnosis and inform their perception on how to guide the patients.

## 2. The Top 5 Things That General Radiologists Need to Know

### 2.1. Definition—Pathogenesis

SLAC is a disorder of the wrist, resulting from altered stress around an unstable scaphoid [1]. SNAC results from a scaphoid fracture that has not healed properly (nonunion fracture). SLAC is a progressive type of wrist OA, more often as a result of a scapholunate ligament (SLL) tear that can be either traumatic or non-traumatic (Table 1).

The SLL is of primary importance in wrist stabilization. It consists of three parts (dorsal, membrane and volar) with the dorsal part being the thickest and strongest. The instability induced by the SLL tear leads to loss of synchronized motion between the scaphoid and lunate bones and gradually results in dorsal intercalated segment instability (DISI), in which the lunate rotates dorsally around its joint with the radius. These changes have an impact on carpal orientation, particularly on the position of the scaphoid, which shifts stress to radioscaphoid and capitolunate joints. As SLAC progresses, the capitate migrates proximally, displacing the lunate ulnar-ward, leading to carpal collapse and midcarpal OA. In severe forms of SLAC, the radiolunate joint is spared because of its spherical

shape. On the contrary, the radioscaphoid joint is prone to degeneration due to its elliptical morphology, which cannot accommodate asymmetrical loading. The radioscaphoid joint motion can be simulated with two nested spoons; when one is rotated by 90°, the axes of the spoons shift and are perpendicular to each other, resulting in the destruction of their articular surface. Rotary subluxation of the scaphoid causes excessive loading to the radioscaphoid articulation, incongruence and contact at the dorsal and volar aspects of the radioscaphoid joint, resulting in a SLAC wrist [1].

**Table 1.** Causes of SLAC [1].

| | | |
|---|---|---|
| 1. | Traumatic | Scapholunate ligament rupture |
| 2. | CPPD [1] | Pyrophosphate leading to SLL [1] tear |
| 3. | Preiser's disease | Idiopathic avascular scaphoid necrosis |
| 4. | Kienböck's disease | Avascular necrosis of the lunate |
| 5. | Intra-articular fractures | Involving the radioscaphoid or capitolunate joint |

[1] SLAC: scapholunate advanced collapse; CPPD: calcium pyrophosphate dihydrate crystal deposition disease; SLL: scapholunate ligament.

SNAC, as firstly described by Vender et al. in 1987 [2], is a cause of wrist OA with a similar pattern to SLAC. In SNAC, as the name suggests, there is a non-union fracture with subsequent avascular necrosis of the scaphoid, but the SLL remains intact [2]. As a result, the proximal scaphoid fragment remains united with the lunate and the distal fragment can move independently. The articulation between this unstrained fragment and the distal radius is affected, and osteoarthritic changes start in the non-union site, which slowly progress to the joints of the wrist, causing dramatic modifications to carpal kinematics. DISI is one of the first alterations that appear in SNAC because proximal scaphoid and lunate are no longer in synchronous motion with the distal scaphoid. Proximal third scaphoid non-union fractures are related to a higher percentage of cartilage destruction and a more severe DISI. The remainder of the degenerative alterations in joint motion in SNAC are similar to SLAC. The two main differences between SLAC and SNAC are that in the latter, the proximal scaphoid is linked to the lunate via the intact scapholunate ligament, and the joint space is usually preserved. Vender et al. (1987) [2,3] defined the stages of SNAC arthritis slightly different as compared to the stages presented by Watson and Ballet [4], which are the ones used currently for both SLAC and SNAC (Table 2). These dissimilarities are crucial because they interfere with the treatment planning in both SLAC and SNAC.

**Table 2.** The two main classifications of SLAC and SNAC.

| Stages | Vender et al. on SNAC | Watson and Ballet on SNAC and SLAC |
|---|---|---|
| Stage I | Involvement of the joint between radial styloid and distal scaphoid fragment | Articulation between radial styloid and scaphoid is involved |
| Stage II | Involvement of the joint among proximal scaphoid fragment and capitate data | Plus, proximal radioscaphoid joint |
| Stage III | Radioscaphoid, scaphoid-capitate and capitolunate articulation is affected | Plus, capitolunate articulation |

## 2.2. Clinical Picture and Physical Examination

Both SLAC and SNAC can be asymptomatic and thus go undiagnosed for many years. Upon commencing, symptoms consist of pain during manual activities, restriction of range of motion (ROM) and reduction in grip strength. For as long as SLAC or SNAC remain untreated because of lack of symptoms or a delay in diagnosis, the ongoing instability results in wrist OA. This detainment may interfere with the treatment choices, and overall, the patient's quality of life due to severe functional impairment. Physical examination may reveal dorsal-radial wrist swelling, reduced ROM, pain and scaphoid tenderness upon compression, radial translation and carpal supination, especially in severe cases [5]. As there are no specific clinical tests, imaging plays a key role in diagnosis, following the progression and evaluating post-treatment changes of SLAC.

## 2.3. Imaging Diagnosis

In wrist OA, radiographs are the first choice with computed tomography (CT) and magnetic resonance imaging (MRI) to follow, depending upon the findings. SLAC and SNAC have a predictable pathway (scapholunate dissociation/diastasis, subluxation and DISI), which eventually leads to carpal collapse and can be evaluated through specific imaging findings. This progression of biomechanical changes has been described in three stages with which every radiologist should be familiar. In Stage I, the radial radioscaphoid joint is narrowed; in Stage II, there is a progression to the entire radioscaphoid joint; and in Stage III, the entire capitolunate articulation is affected.

### 2.3.1. Radiographs

Radiographs are the first-line, low-cost and widely available imaging to be obtained when SLAC or SNAC is suspected. They can evaluate the osseous structures, bone alignment and joint spaces and are a useful tool for the diagnosis of osteoarthritis (OA) provided that the radiographic technique is optimum. Early stages show normal joint space (Figures 1 and 2). Radiographic changes are only manifested when wrist OA is well established. Thus, the joint space may be normal in the presence of severe symptoms as radiographs do not directly depict the articular cartilage [6].

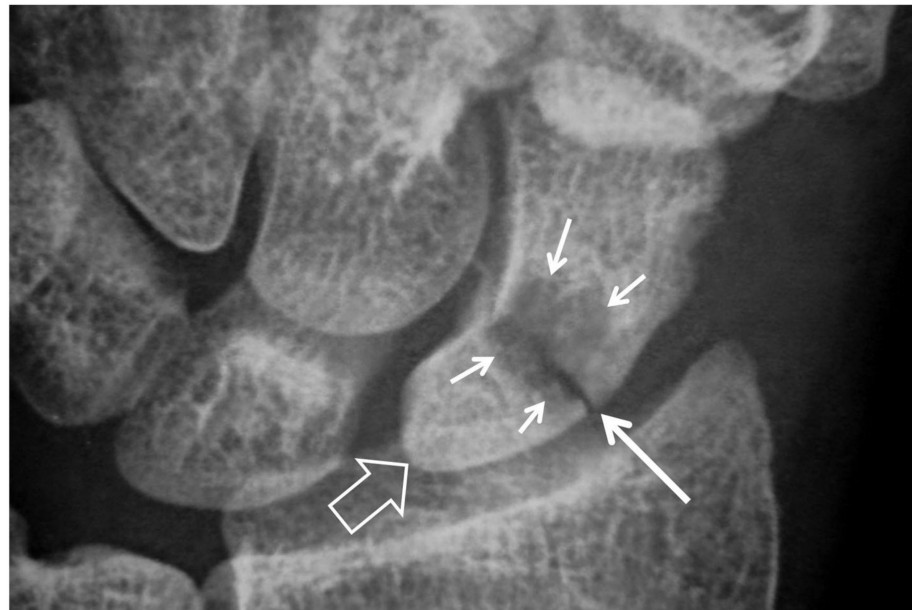

**Figure 1.** A 23-year-old male patient with a history of a scaphoid fracture 1 year prior to current imaging. PA radiograph showing pseudarthrosis (arrow), proximal pole osteosclerosis in keeping with osteonecrosis (open arrow) and cyst formation on both sides of the previous fracture (short arrows). The radioscaphoid joint space is intact.

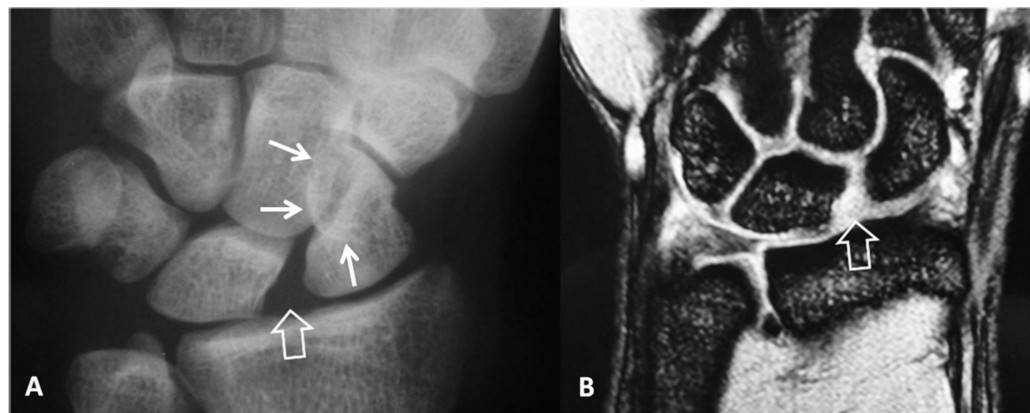

**Figure 2.** A 34-year-old orthopedic surgeon, practicing in Boxing, presents with 3-month period of pain and limited range of motion.(**A**) PA radiograph showing the increased scapholutane distance suggesting scapholunate ligament rupture (open arrow). The finding is often called as the "Thery-Thomas" sing, after the famous actor teeth appearance. The "signet ring" sign is also evident due to the foreshortening of the scaphoid (arrows). (**B**) Coronal T2* MR image showing the absence of the low signal intensity membranous ligament (open arrow) in keeping with scapholunate ligament tear. The radioscaphoid joint space is intact.

The three stages of SLAC can be distinguished on radiographs. In Stage I, there is damage only to the most radial aspect of the radioscaphoid joint and the radial styloid process loses its normal shape. In Stage II, narrowing of the whole radioscaphoid joint occurs, and in Stage III, there is in addition narrowing of the capitolunate joint and proximal migration of the capitate with ulnar-ward transposition of the lunate (Figure 3).

One of the imaging findings of SLAC and SNAC wrist is the rotary subluxation of the scaphoid. This is defined as dorsal scaphoid displacement with rotation onto the dorsal side of the radius. This can be presented on the PA radiograph as scapholunate diastasis—often described as the "Terry Thomas" sign (referring to the gap the actor had in his teeth) (Figures 2 and 4). Scapholunate diastasis is an abnormal increase in the scapholunate interval, which occurs when there is a complete tear of the SLL and is the result of scapholunate dissociation, which is the loss of synchronous motion of scaphoid and lunate. Another finding is the foreshortening of the scaphoid, often referred as the "signet ring sign" (Figure 2). In the lateral radiograph, there is an increased radioscaphoid angle of >60°, a scapholunate angle of >60°–80°and a normal radiolunate angle of 30°–60°. Subluxation of the scaphoid onto the dorsal rim of the radius can also be seen in lateral radiographs but is sometimes difficult to assess.

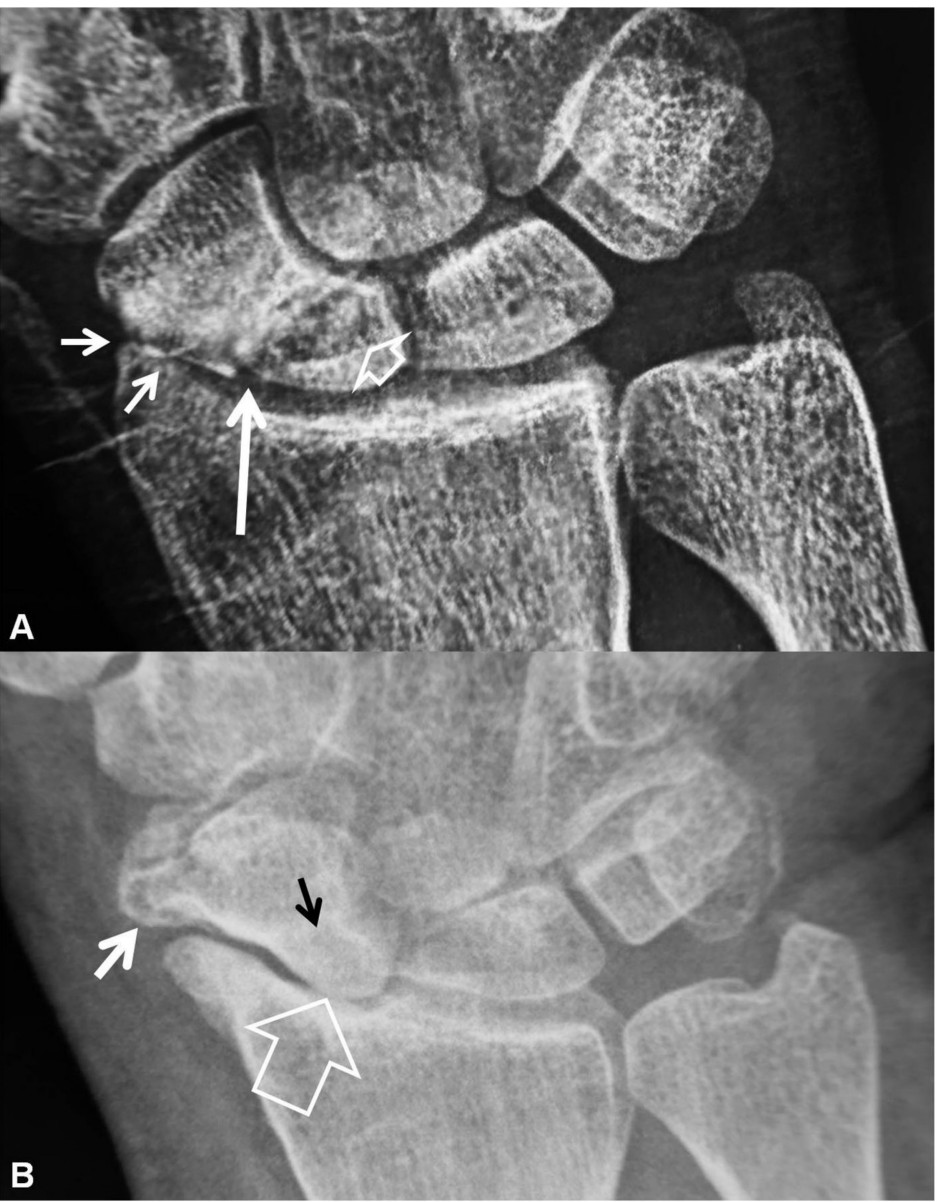

**Figure 3.** PA radiographs. (**A**) A 35-year-old male patient with a history of wrist pain for 10 years. Scaphoid fracture non-union (arrow), proximal pole cyst formation (open arrow) and radial side joint space narrowing (short arrows) are diagnostic of stage I SNAC wrist. (**B**) A 44-year-old male patient known scaphoid osteonecrosis. Proximal pole osteonecrosis (black arrow) combined with radioscaphoid space narrowing (open arrow) and osteophytosis of the scaphoid bone (short white arrow) are diagnostic of a SNAC wrist. The proximal carpal arc is disrupted and the capitate bone is proximally migrated.

Concerning DISI, the imaging findings in lateral radiographs are a radiolunate angle of >10°, a capitolunate angle of >30° and/or a scapholunate angle of >80° [7]. Particularly for SNAC, radiographs can also detect the non-union scaphoid fracture and the induced avascular necrosis (AVN) of the proximal scaphoid fragment (Figure 5). Knowing the vascular supply to the scaphoid is valuable in understanding AVN. The primary internal vascular supply enters the scaphoid dorsally (at 70%) mainly to its distal pole. The remaining 30% enters the scaphoid palmary, also through the distal pole. Since the scaphoid's predominant blood supply has a distal to proximal orientation, the proximal scaphoid fragment may disrupt the blood supply and end up at the AVN of the scaphoid. AVN is associated with the position of the fracture, with the proximal fragments having a 100% chance of

AVN. The primary radiographic finding of AVN of the proximal scaphoid fragment is its irregular contour with sclerotic and lytic lesions. The rest of DISI and OA findings in the radioscaphoid joint may also be present in SNAC similar to the way they are presented in SLAC [8].

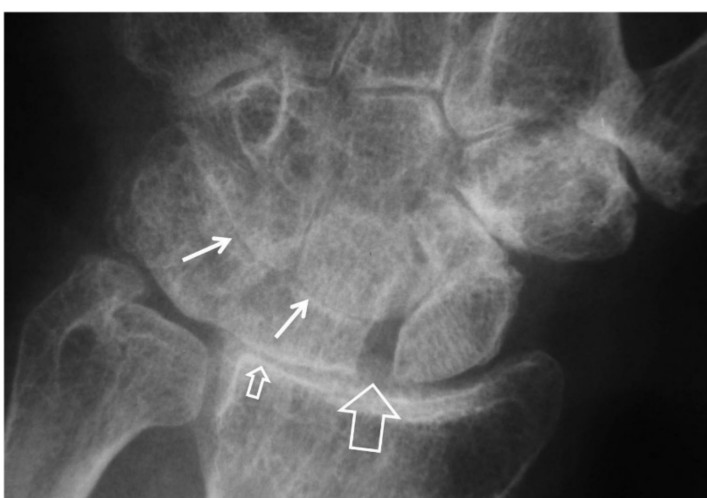

**Figure 4.** A 55-year-old male patient with long standing wrist pain and limited range of motion. Scapholunate ligament disruption (large open arrow), radiolunate joint space narrowing (small open arrow) and mid-carpal osteoarthritis (arrows) are diagnostic of a SLAC wrist. The altered shape of the scaphoid is due to its rotatory subluxation.

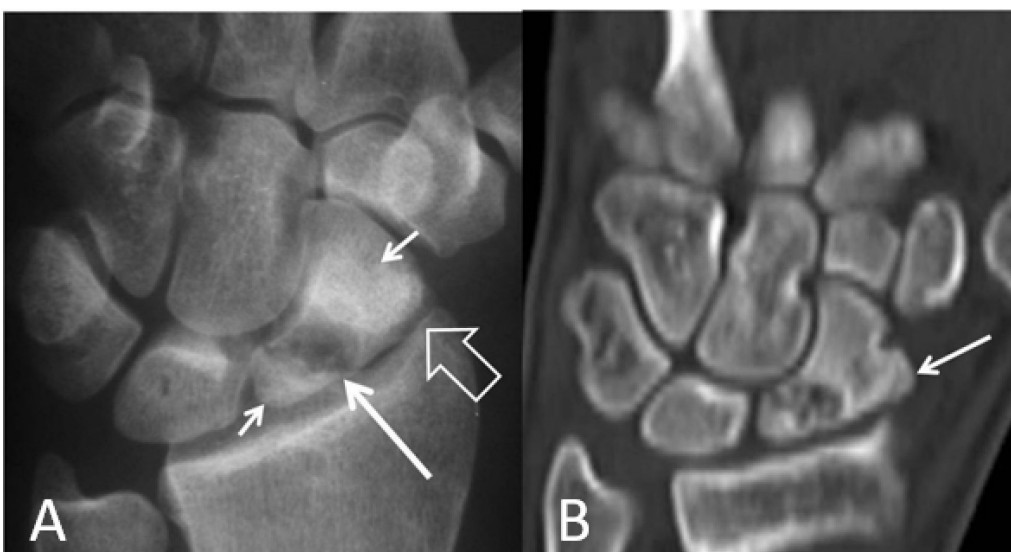

**Figure 5.** Scaphoid pseudarthrosis and early SNAC wrist, in this 25-year-old male patient with an injury 4 years prior to current imaging. PA radiograph (**A**) showing the pseudarthrosis (long arrow), proximal and distal pole osteonecrosis (small arrows) and joint space narrowing at the most radial aspect of the radioscaphoid joint (open arrow). Coronal CT reconstruction (**B**), showing in addition osteophyte formation of the scaphoid bone (arrow).

### 2.3.2. Computed Tomography/CT-Arthrography

Multidetector computed tomography (MDCT) is a valuable tool in evaluating the wrist's joint spaces, because of its high spatial resolution and the absence of overlapping tissues. Plain MDCT images may show the early degenerative findings, including subchondral bone sclerosis with or without cyst formation (Figures 5–7).

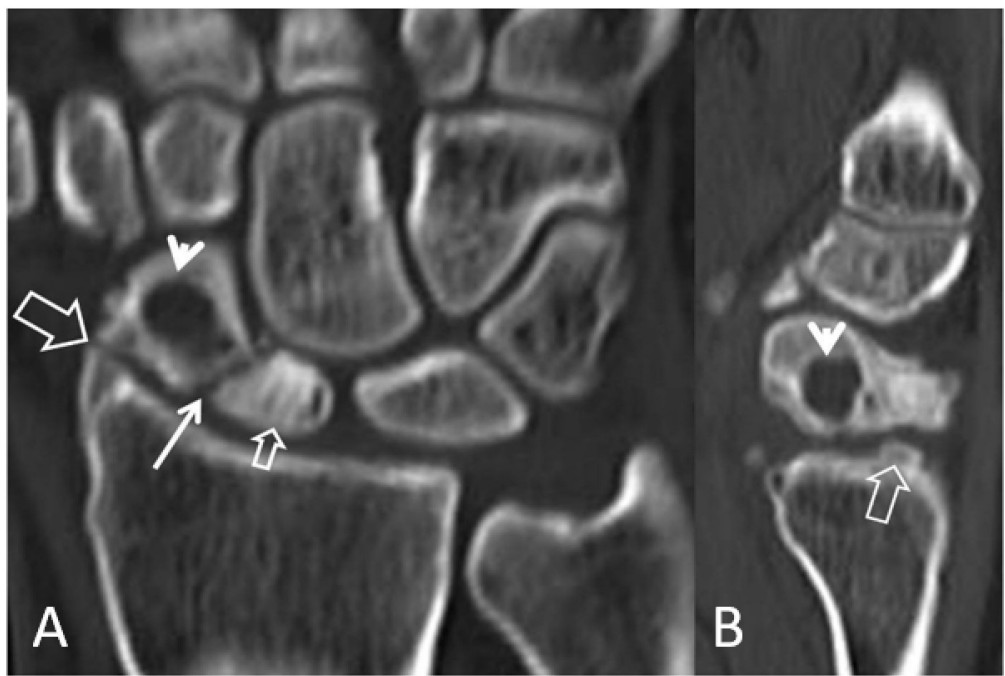

**Figure 6.** Coronal (**A**) and sagittal (**B**) CT reconstructions of a 23-year-old male patient with an injury 7 years prior to current imaging. SNAC wrist is shown with radioscaphoid joint space narrowing with osteophyte formation (large open arrow left, open arrow right), secondary to proximal pole osteonecrosis (small open arrow left) and pseudarthrosis (thin arrow left). Large cyst formation is shown in the distal pole of the scaphoid (arrowheads).

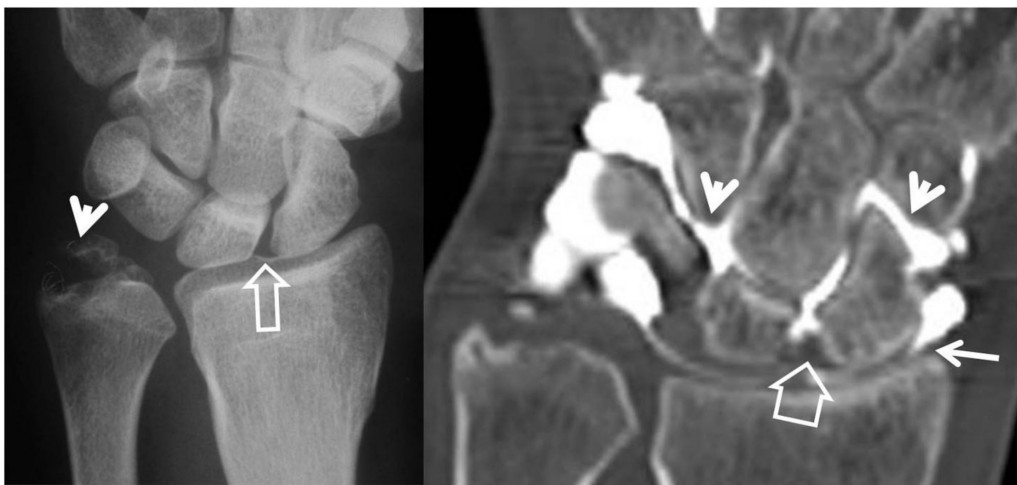

**Figure 7.** A 24-year-old female athlete with a surgically proven scapholunate ligament disruption. PA radiograph (**left**) image shows scapholunate diastasis (open arrow) and an old fracture of the ulnar styloid (arrowhead). Coronal MDCT arthrographic image (**right**) shows midcarpal joint opacification (arrowheads) although the SLL looks intact (open arrow). The injection site was distal to the radial styloid (thin arrow). A small dorsal SLL tear was found in the theatre.

CT-arthrography (CTa) is performed with an iodine contrast material injected in specific wrist joints. For depicting a SLL tear, a single injection at the radioscaphoid joint is performed in our department under fluoroscopic or ultrasound guidance. In the presence of a tear, the midcarpal joints will be opacified (Figure 8).

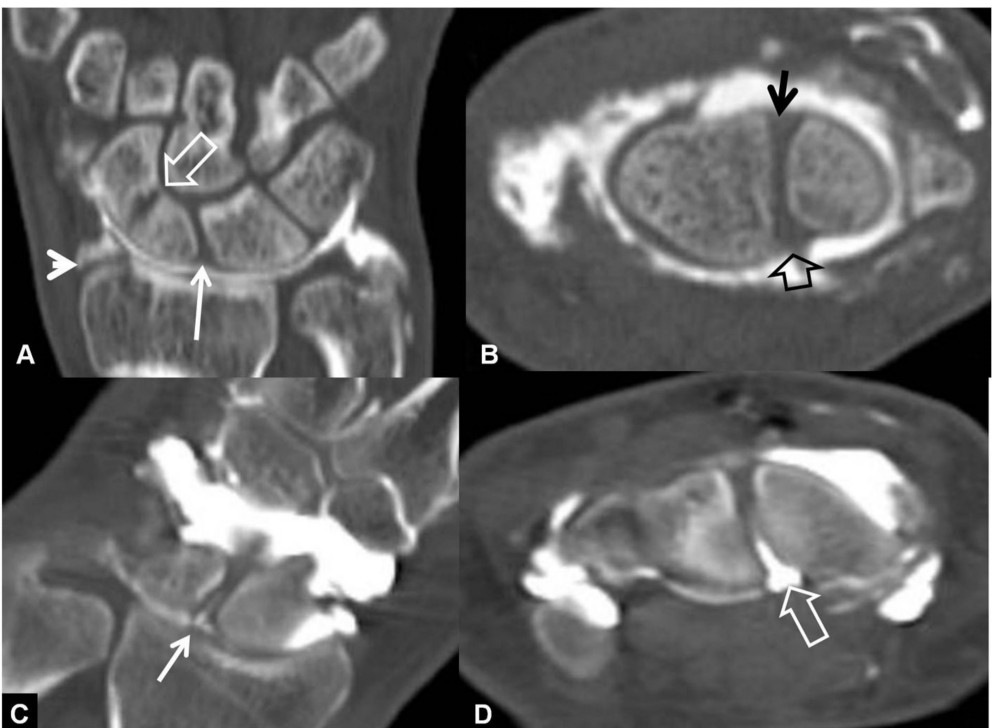

**Figure 8.** A 33-year-old with a history of scaphoid fracture. (**A**) Coronal CT-arthrographic image showing the scaphoid pseudarthrosis (open arrow), a radial styloid process fracture (arrowhead) and an intact scapholunate ligament (arrow). (**B**) Axial CT-arthrographic image showing the intact dorsal (arrow) and palmar (open arrow) parts of the scapholunate ligament. A 56-year-old male patient who was involved in a motor vehicle accident 1 year prior to current imaging. Coronal (**C**) and axial (**D**) CT-arthrographic images showing a tear at the membranous (arrow) and palmar (open arrow) parts of the scapholunate ligament.

In SNAC, one of the main MDCT findings is the exact location of the non-union scaphoid fracture (Figures 5b, 6a, 7 and 9). This location is important for the prognosis of the disease as a non-union fracture at the proximal third of the scaphoid has a higher chance of dorsal flexion of the lunate and is thus more prone to extensive cartilage degeneration.

In SLAC, one of the main CTa findings is the partial or complete tear of SLL [9,10]. Axial images are more suitable for the estimation of the volar and dorsal part of SLL, the coronal for the membranous part of SLL and sagittal reconstruction images are appropriate for the evaluation of DISI (Figure 9).

In CT, as with radiographs, OA changes of SLAC and SNAC, as a result of scaphoid rotary subluxation and DISI, include the increase of the scapholunate angle (>60°) and the capitolunate angle (>30°), provided that the hand lies in a neutral position.

The drawbacks of CT imaging are the radiation burden, the inability to directly image the cartilage degeneration without injecting contrast into the joints, its limitation in the evaluation of soft tissue irregularities and the lack of standardized interpretation criteria of the findings.

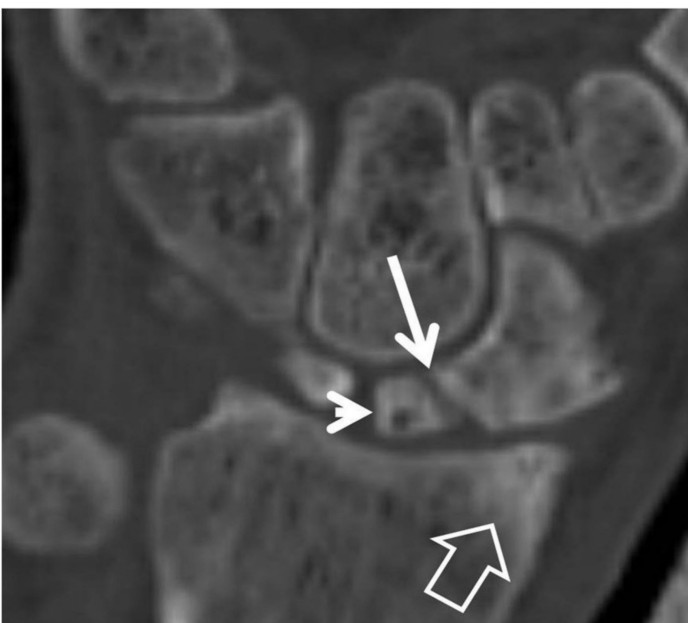

**Figure 9.** A 42-year-old male patient with pseudarthrosis following previous scaphoid fracture. Coronal CT reconstruction showing the fracture non-union (arrow), the proximal pole osteosclerosis with cyst formation (arrowhead) and the subarticular osteosclerosis of the radial styloid (open arrow) in keeping with early osteoarthritis.

### 2.3.3. Magnetic Resonance Imaging/MRI-Arthrography

Magnetic resonance imaging (MRI) and MRI-arthrography (MRa) should be included in the diagnosis of SLAC and SNAC as they provide superb imaging of the osseous and soft tissue structures. In addition, MRI is a robust approach for the evaluation of carpal OA as it can directly evaluate the bone marrow and the osteoarticular deformities (Figure 10). Moreover, MRI can evaluate the vascularity of the bone marrow by estimating the degree of enhancement (Figure 11).

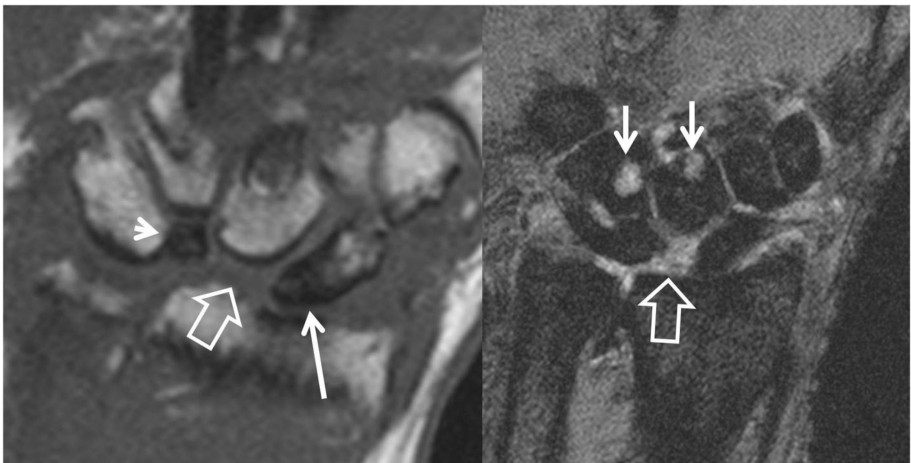

**Figure 10.** A 56-year-old male patient with SLAC and SNAC wrist. Coronal T1w (**left**) and T2* (**right**) MR images, showing the proximal pole scaphoid necrosis (arrow), the scapholunate diastasis with proximal capitate bone migration (open arrow), the rotation of the lunate bone (arrowhead) and the subchondral cysts secondary to osteoarthritis development (short arrows).

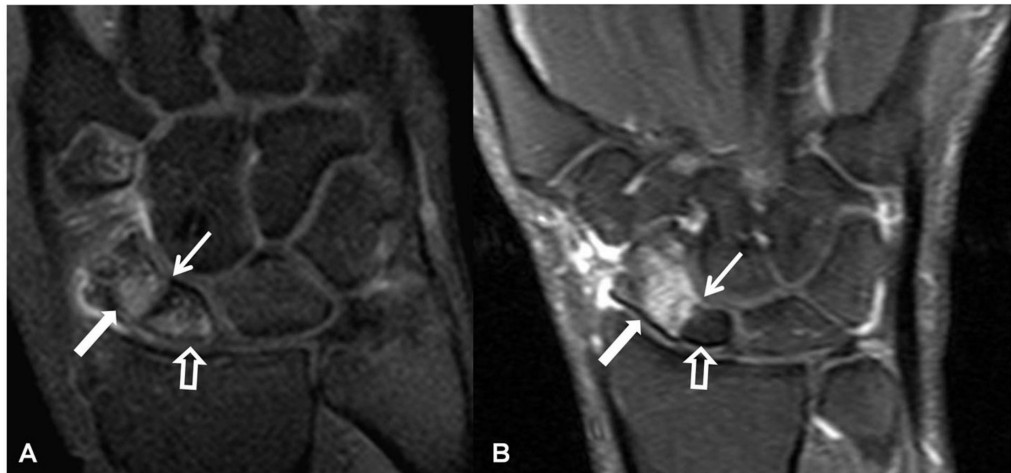

**Figure 11.** Two patients with scaphoid fracture non-union, fat suppressed contrast enhanced T1w coronal MR images. Both images show the fracture line (thin arrows) and the normal enhancement following a fracture, of the distal pole of the scaphoid bones (thick arrows). (**A**) Enhancement of the proximal pole suggests viable bone marrow (open arrow). (**B**) Absence of any enhancement of the proximal pole suggests nonviable bone marrow (open arrow). The signal intensity of the proximal pole is lower as compared to the normal bone marrow of the surrounding bones.

MRa can accurately assess the SLL disruption by depicting the absence of the ligament and the opacification of the midcarpal joints (Figure 12) [11]. Early changes of the articular cartilage in extreme detail concerning depth, height and thickness can be depicted (Figure 13) [12]. Axial images on MRI and MRa are more suitable for the estimation of wrist ligaments and sagittal images for the evaluation of scaphoid bone and DISI (Figure 14). Currently, MRI and MRa are the most widely preferred methods in diagnosing wrist pain [13]. Comparison of the various imaging modalities is shown in Table 3.

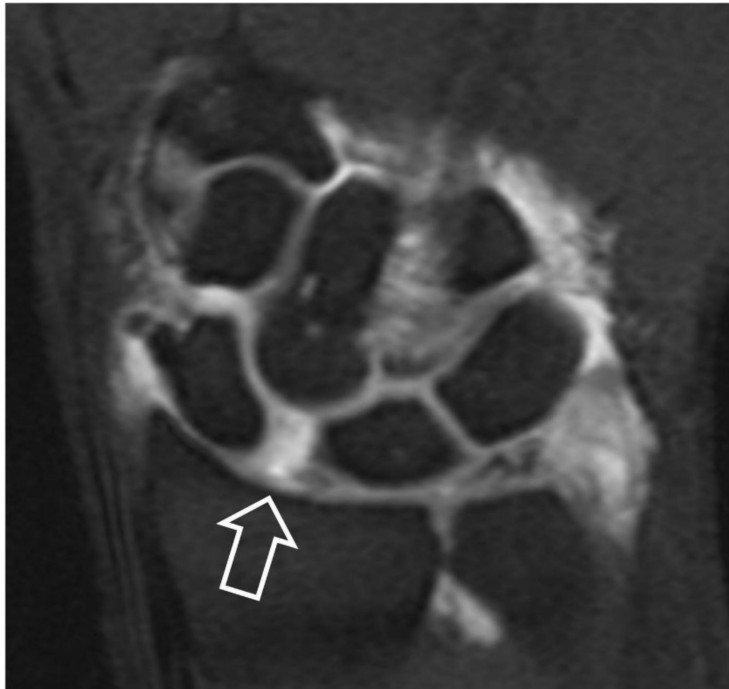

**Figure 12.** Coronal fat suppressed T1w MR arthrographic image following a single injection in the radiocarpal joint. Scapholunate diastasis (open arrow) and opacification of the midcarpal joints suggest SLL disruption.

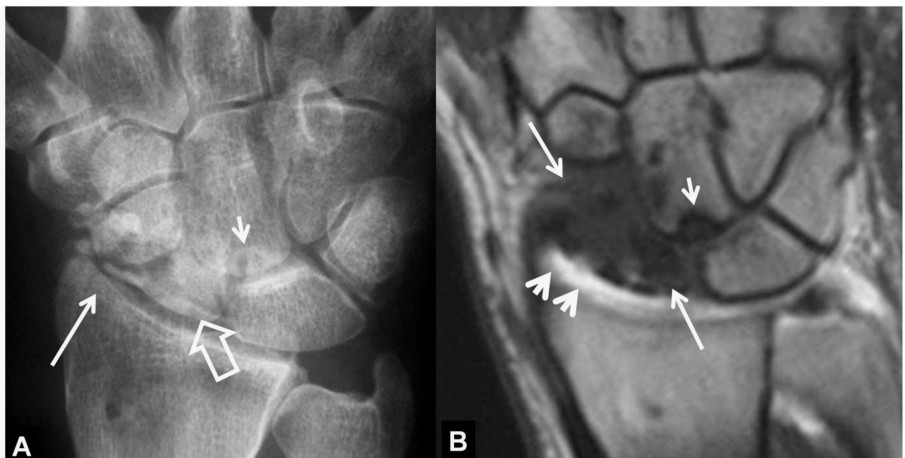

**Figure 13.** A 41-year-old male patient with a scaphoid fracture 5 years prior to current imaging and persistent pain and functional impairment during the last 2 years. (**A**) PA radiograph showing proximal pole osteosclerosis in keeping with osteonecrosis (open arrow) and radial radioscaphoid joint space narrowing with minimal subchondral sclerosis in keeping with SNAC stage I disease (arrow). (**B**) Coronal T1w MR arthrographic image showing irregularity of the radioscaphoid space (arrowheads) and low signal intensity of the scaphoid bone marrow suggesting global osteonecrosis (arrows). Subchondral sclerosis suggesting early osteoarthritis is shown on both images in the proximal capitate (small arrows).

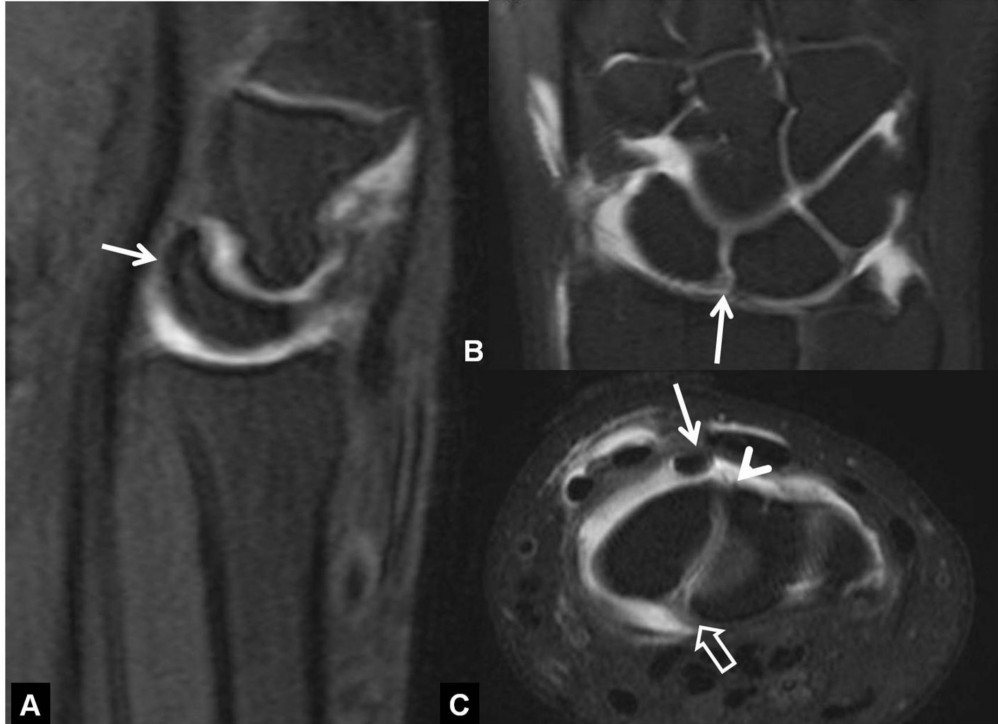

**Figure 14.** DISI in a 66-year-old-male patient with SLAC wrist. (**A**) Sagittal MR arthrographic fat suppressed T1-w image showing the dorsal rotation of the lunate (arrows). (**B**) Coronal image showing a partial tear of the membranous part of the SLL. (**C**) Axial image showing the intact dorsal (arrowhead) and palmar (open arrow) parts of the SLL. The long arrow shows an air bubble injected with the contrast.

**Table 3.** Comparison of imaging modalities on SLAC and SNAC.

| Comparison | Radiographs | CT-CTA | MRI-MRa |
|:---:|:---:|:---:|:---:|
| PROS | Widely available | Evaluates cartilage, ligament lesions | Depicts osseous/soft |
| | Low-cost | Spatial ability | Evaluates bone marrow |
| | | Quick-minimal motion artifacts | Estimates fracture time |
| | Detects established osteoarthritis | Post-operative monitoring Estimates bone fractures | Assesses ligaments, cartilage and subchondral lesions |
| CONS | Dorsal displacement of the scaphoid is difficult to assess | No standardized interpretation | Time-consuming/motion artifacts |
| | Osteoarthritic changes are obvious only when they are well established | | Partial tear of SLL not well detected |
| | Only indirectly detects cartilage damage via space joint narrowing | Radiation | Not feasible in the presence of metallic objects |

## 2.4. Differential Diagnosis

SLAC and SNAC should be distinguished from disorders that may have similar clinical and radiological findings as they have a different progression, prognosis and treatment.

Calcium pyrophosphate deposition or CPPD) has been described as a should be cause of wrist OA (Figure 15). As the prognosis and treatment of pseudogout may differ, the distinction between the two disorders is crucial. CPPD, contrary to SLAC, has a non-traumatic etiology, affects multiple joints and is predominantly bilateral. Romano (2003) [14] described a four-stage pattern of the scaphoid chondrocalcinosis advanced collapse (SCAC). In Stage I, only the radioscaphoid joint is affected; in Stage II, there is scapholunate dissociation and narrowed capitolunate joint; in Stage III, the difference with SLAC and SNAC is more obvious as the scaphoid is impacted into the distal radius; in Stage IV, there is a pan-carpal arthritis as STT, where triquetrolunate, radiolunate and midcarpal joints are affected. In SCAC, the scaphoid is described as extended, contrary to flexed, which is described in SLAC [15]. Preiser's disease in Figure 15, was an incidental finding and may have been the result of autoimmune hemolytic anemia, systemic lupus erythematosus and renal transplantation.

Another entity that should be differentiated is STT OA. About 25% of STT OA cases is related to DISI, with the difference that there is no involvement of the proximal carpal row. Contrary to SLAC and SNAC, the radioscaphoid articulation is spared.

Lastly, SNAC and SLAC should be differentiated from Kienböck's disease. The rotary subluxation of the scaphoid that occurs in Kienböck's disease does not lead to SLAC [16]. When chronic subluxation is present, there is an alteration in the articular radioscaphoid surface and the radioscaphoid space joint is spared.

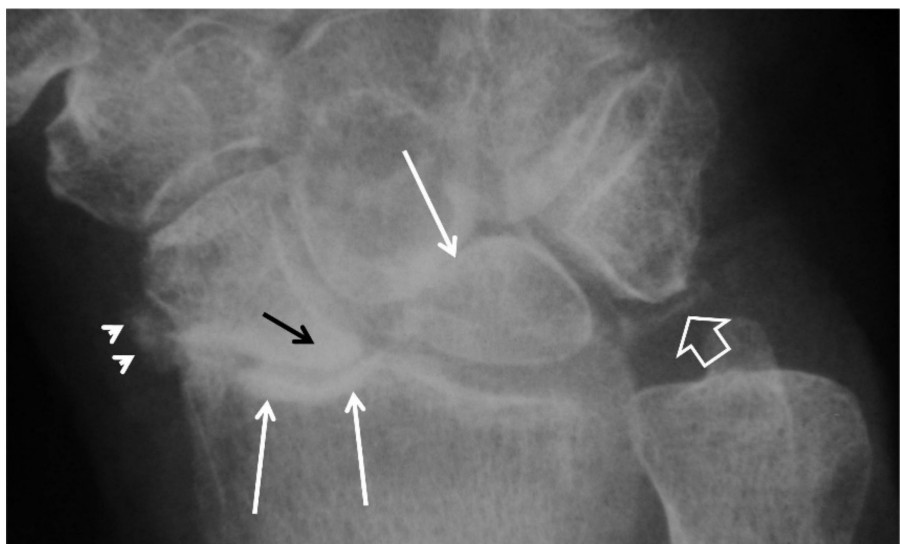

**Figure 15.** A 62-year-old male patient, with chronic wrist pain. PA radiograph showing scaphoid proximal pole osteosclerosis, in keeping with Preiser's disease (black arrow). Chondrocalcinosis is shown in the triangular fibrocartilage in keeping with CPPD (open arrow) along with soft tissue calcifications radially (arrowhead). Osteoarthritic changes are evident in the radioscaphoid joint space with collapse of the articular surface and the midcarpal joints with proximal migration of the capitate (long white arrows).

*2.5. Treatment*

Wrist OA is one of the main reasons for wrist pain and is characterized by degenerative changes leading to swelling, loss of motion and decrease in grip strength. Therefore, the treatment of SLAC and SNAC has a significant impact on the patient's quality of life.

Treatment of SLAC and SNAC varies depending on their lesion stage and patient symptoms. Initially, in asymptomatic SLAC, there is no suggested treatment. For symptomatic SLAC and SNAC cases, the first-line therapy is conservative. Nonsteroidal anti-inflammatory drugs, wrist splints and casts, motion restriction and intra-articular corticosteroid injections are options that can also be used in patients who are not suitable for surgery. Intra-articular injections under x-ray fluoroscopy or ultrasonography are gaining ground as minimally invasive procedures that can retard surgery. Corticosteroids and hyaluronic acid injections are commonly used in wrist OA with satisfactory results. Amniotic and umbilical cord (AMUC) [17] and platelet-rich plasma (PRP) injections have also been reported but are still in early stages of research and development [18].

When all the previously described measures are not effective and symptoms are exacerbated or the diagnosis is made at a later stage, surgery is the next step. Surgical alternatives for SLAC and SNAC are proximal row carpectomy, arthrodesis, partial or complete (four-corner arthrodesis), radial styloidectomy and total wrist denervation, while SNAC has one extra option, with the removal of the distal non-united scaphoid fragment [19] (Table 4).

**Table 4.** Treatment of SLAC and SNAC.

| Stage I | Stage II | Stage III |
|---|---|---|
| Radial styloidectomy | Proximal row carpectomy | Proximal row carpectomy |
| Scapholunate reconstruction | Midcarpal arthrodesis | Four corner arthrodesis |
| Distal scaphoid excision | | Total wrist denervation |

In Stage I, the common treatment approaches are radial styloidectomy with or without scaphoid or scapholunate reconstruction and excision of the distal scaphoid. When

radial styloidectomy is performed alone, it is not capable of preventing the osteoarthritic changes in the wrist and the symptoms usually recur. Additionally, with the excision of the scaphoid distal pole, which has been used in Stage I SNAC and SLAC, there is a chance of provoking further wrist destabilization, despite its temporarily positive outcomes. As a result, the most widely preferred method is a combination of radial styloidectomy with scaphoid grafting.

In Stage II, proximal row carpectomy (PRC) and midcarpal arthrodesis (MCA) are the two main procedures for SLAC and SNAC with partial joint replacement being applied, but these two procedures sometimes display poor outcomes. PRC requires the removal of the scaphoid, lunate and triquetrum, and hence an articulation among the capitate and the lunate fossa of the radius is created [20]. The main contraindication for PRC is the eroded cartilage of the two remaining surfaces. To overcome this obstacle, partial joint replacement can contribute, by resurfacing the head of the capitate. The advantages of PRC are fast mobilization and pain relief at a satisfactory level and disadvantages are the shortening of the carpus, poor grip strength, the short-term results (especially for younger patients) and the likelihood of radiocapitate OA. Conversely, MCA, which is the fusion of the scaphoid, lunate, triquetrum, capitate and hamate, maintains carpal height, provides pain relief, a sufficient range of motion (ROM) and a certain grip strength, but carries the risk of non-union [21].

MCA, PRC and four-corner fusion are options for the treatment of Stage III SLAC or SNAC wrist. Four-corner arthrodesis (FBA) includes the excision of scaphoid and the fusion of the capitate, lunate, triquetrum, and hamate bones with the use of an implant [22]. As the capitate is part of the four-bone fusion, four-corner arthrodesis could be performed when the capitate is affected, making FBA possible, including in the last stage of SLAC and SNAC. Through the years, several implants have been used in FBA, such as K-wire, Herbert screws and specific plates, with locking plates being the most recently introduced method [23]. The ultimate objective of all these trials was to improve carpal stability, final ROM and to provide early mobilization when this method is used. Regarding pain relief, FBA and PRC—which is the next most frequently used procedure in Stage III—have equally sufficient outcomes, but PRC provides better ROM and does not require an implant. On the contrary, FBA provokes fewer degenerative impairments, maintains normal carpal height and provides better grip strength [24].

Total wrist denervation is also an option, especially for late disease stages, but has also been used at any stage. Denervation can be either partial (only posterior or interior interosseous nerves) or complete (all branches). Total wrist denervation is combined with poor morbidity rates and adequate pain relief and ROM. The two main categories of patients recommended to undergo this procedure are light workers who have a sufficient ROM, or elderly patients. Thus, total wrist denervation has a significant impact on pain relief through the removal of pain signals, but it does not address the etiology of SLAC and SNAC.

The appropriate surgical methods are selected depending on osteoarthritis changes, age, range of motion and the patient's quality of life (job, manual demands). SLAC and SNAC's treatment methods continue to evolve, and new techniques are being developed with promising clinical and radiological results [25].

## 3. Conclusions

SLAC and SNAC are the most common types of degenerative arthritis in the wrist. Absence of symptoms is possible, but mechanical pain, limitation of motion range and decreased strength commonly affect patients' quality of life and make early diagnosis essential. Imaging is a valuable tool alongside the clinical examination. Radiographs as an easy and low-cost modality is the first line of the diagnostic approach. CTa and MRa are also useful for a more detailed examination, and they are valuable preoperative and postoperative tools. Lastly, treatment approach has a significant impact on wrist

functionality and if wisely chosen, can truly affect patients' life, providing satisfactory wrist mobility and a pain-tolerable or pain-free life.

**Author Contributions:** Conceptualization, A.H.K.; methodology, A.H.K.; data curation, E.K. and M.P.; writing—original draft preparation, E.K. and M.P.; writing—review and editing, A.H.K. All authors have read and agreed to the published version of the manuscript.

**Funding:** This research received no external funding.

**Institutional Review Board Statement:** Not applicable. Ethical approval is waived because this is a review article not involving humans or animals on experimental study.

**Informed Consent Statement:** Not applicable.

**Conflicts of Interest:** The authors declare no conflict of interest.

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
