# Peer review of "SLAC and SNAC Wrist: The Top Five Things That General Radiologists Need to Know"

_tomography, doi:10.3390/tomography7040042_

Round 1

Reviewer 1 Report

Thank you for the opportunity to review this manuscript. This narrative review finding in SLAC and SNAC wrist. 

My major recommendation is more optimal use of references. There are statements though out the manuscript which are without cited references.

Please check for typographical errors.

Comments:

Line 14 - Please rephrase. Wrist OA is seen with SLAC/SNAC but does not only occur with these conditions.

Line 26 - Please provide reference for this statistic (95%).

Line 44—59  Need references for the statements in this paragraph

Figure 2- Picture of Terry Thomas should not be used (besides copyrights violation, this is not appropriate for the article). 

Line 114 “sign” is incorrectly spelled. 

Line 203 - degeneration may be better than “erosion” 

Author Response

Line 14 - Please rephrase. Wrist OA is seen with SLAC/SNAC but does not only occur with these conditions.

We agree with the reviewer's comment. In the introduction we suggest that " Scapholunate advanced collapse (SLAC) and scaphoid non-union advanced collapse (SNAC) are the most common patterns (55%) of wrist OA, which follow a predictable sequence of degenerative changes." The rest 45% is obviously the result of other conditions. However, we modified slightly the text in the abstract in order to further clarify the statement: Wrist OA often results from scapholunate advanced collapse (SLAC) or scaphoid nonunion advanced collapse (SNAC).

Line 26 - Please provide a reference for this statistic (95%).

Wrist osteoarthritis (OA) occurs almost exclusively (95%) as a periscaphoid problem and it can result in severe functional disability

now reads as:

Wrist osteoarthritis (OA) occurs in about 95% as a periscaphoid problem and it can result in severe functional disability [1].

Line 44—59  Need references for the statements in this paragraph

Done

Figure 2- Picture of Terry Thomas should not be used (besides copyrights violation, this is not appropriate for the article). 

The picture is widely available on the web and there were no copyright issues mentioned. In addition, the picture was modified with photoshop.

Line 114 “sign” is incorrectly spelled. 

Corrected

Line 203 - degeneration may be better than “erosion” 

the word "erosion" now reads as "degeneration"

Reviewer 2 Report

SLAC and SNAC Wrist: The Top Five Things That General 2 Radiologists Need to Know

Comments

Wrist osteoarthritis (OA) can be the preface of a long list of skeletal and extra-skeletal pathologies. Therefore, it is mandatory that Authors must connect the so many images they presented with the differential diagnosis. Radiologists need to know how to connect wrist OA with other skeletal and extra-skeletal heritable or non-heritable disorders. Otherwise, segmental identification is of little clinical significance. 

For Preiser disease or avascular necrosis (AVN) of the scaphoid, this is mostly a symptom complex rather than a diagnosis. It can be encountered in patients with autoimmune hemolytic anemia, systemic lupus erythematosus, or renal transplantation, or have had other risk factors, such as metabolic syndromes. Also Kienbock’s disease can be seen in carpal tunnel syndrome and other osseous, metabolic and neurogenic disorders.

Figure 1. Apparently a patient with osteoporosis.

Figure 3 is the radiograph of the same patient of figure 1, signifies progressive osteoporosis.

NB: Osteoporosis per se is not a diagnostic entity, it is mostly a symptom complex stemmed from dozens of pathological reasons such as heritable osseous, myogenic, neurogenic, hormonal and other etiologies.

Figure 4. The radiograph of a 55-year-old male patient with long standing wrist pain and limited range of motion most likely in connection with metabolic syndrome such as gouty arthritis.

Figures 5 and 6 showed gradual resorption of lytic changes of the carpal bones, which might reflect osteolytic changes.

All radiographs need to be connected with the diagnosis or differential diagnosis. 

Author Response

Wrist osteoarthritis (OA) can be the preface of a long list of skeletal and extra-skeletal pathologies. Therefore, it is mandatory that Authors must connect the so many images they presented with the differential diagnosis. Radiologists need to know how to connect wrist OA with other skeletal and extra-skeletal heritable or non-heritable disorders. Otherwise, segmental identification is of little clinical significance. 

Generally speaking, OA should be interpreted in the context of differentials. However, extensive discussion as such, in the legend, may be misleading to the readers.

For Preiser disease or avascular necrosis (AVN) of the scaphoid, this is mostly a symptom complex rather than a diagnosis. It can be encountered in patients with autoimmune hemolytic anemia, systemic lupus erythematosus, or renal transplantation, or have had other risk factors, such as metabolic syndromes. Also Kienbock’s disease can be seen in carpal tunnel syndrome and other osseous, metabolic and neurogenic disorders.

I agree, but such a discussion is out of the scope of the article.

Figure 1. Apparently a patient with osteoporosis.

No, definitely not. This is a 23-year-old athletic patient with normal bone density. Osteoporosis cannot be diagnosed on radiographs, with the exception of spine and hip when not traumatic or low-energy fractures occur. The correct nomenclature anyhow is osteopenia, which does not exist in our case.

Figure 3 is the radiograph of the same patient of figure 1, which signifies progressive osteoporosis.

No, definitely not. These are two different and older patients, contralateral wrist.

NB: Osteoporosis per se is not a diagnostic entity, it is mostly a symptom complex stemmed from dozens of pathological reasons such as heritable osseous, myogenic, neurogenic, hormonal, and other etiologies.

I agree, but such a discussion is out of the scope of the article.

Figure 4. The radiograph of a 55-year-old male patient with long-standing wrist pain and limited range of motion was most likely in connection with metabolic syndrome such as gouty arthritis.

No, definitely not. There was no history of gout arthritis or other metabolic disorder. It's just a SLAC wrist.

Figures 5 and 6 showed gradual resorption of lytic changes of the carpal bones, which might reflect osteolytic changes.

I agree that these are osteolytic changes but in the literature, they are described as cysts secondary to fractures.

All radiographs need to be connected with the diagnosis or differential diagnosis. 

This is out of the scope of the article.

Reviewer 3 Report

1. Numerous grammatical errors which need to be addressed.

2. Too much stress was given to the clinical aspects of presentation and can be shortened. Please try to reduce clinical part from the manuscript.

3. Please provide better quality images for CT and MR arthrograms and also provide high-quality reformats of CT scans (figure 7 through 10 and 12 through 14).

4. I particular enjoyed reading treatment related to different stages of SLAC/SNAC.

5. Although the article doesn't provide any additional information on SNAC and SLAC wrists which extensively published previously, I can still see value of the article as a pictorial review.

Author Response

  1. Numerous grammatical errors need to be addressed.

Thorough editing towards correcting the grammatical errors has been done.

  1. Too much stress was given to the clinical aspects of presentation and can be shortened. Please try to reduce the clinical part of the manuscript.

    Indeed, but this was on purpose. This was the aim of the pictorial essay; to present the 5 most important things that clinical radiologists should know on the topic, without extra need for studying the orthopedic literature.

  2. Please provide better quality images for CT and MR arthrograms and also provide high-quality reformats of CT scans (figure 7 through 10 and 12 through 14).

Given the difficulty of collecting a gallery of cases, the cases might be as old as from 2 decades. Thus, it is not easy to replace them with better ones. However, all of them meet the criteria for publishing (>300dpi).

  1. I particularly enjoyed reading treatment related to different stages of SLAC/SNAC.

Thank you, we highly appreciate your positive comment.

  1. Although the article doesn't provide any additional information on SNAC and SLAC wrists which were extensively published previously, I can still see the value of the article as a pictorial review.

Thank you, we highly appreciate your positive comment.

Round 2

Reviewer 1 Report

I again would like to respectfully request that the image of Terry-Thomas is removed. People can do a web search and see these if they want. Therefore it does not need to be in the article. Terry-Thomas sign is commonly mentioned in scientific literature, but it is very unusual to have his pictures in the articles. 

Author Response

We removed the image of Terry-Thomas as suggested. A new image has been uploaded.

Reviewer 2 Report

Articles dealing with skeletal abnormalities should be guided into one Target. This target is the ETIOLOGY UNDERSTANDING. Readers are fade up from the policy of segmental and partitioned medicine. Author responded by saying this is out of the scope of the article. I would like to tell him, there is a big difference between carpeneters who are aiming to fix a borken leg of a chair, and expert physician. Expert physician deals with the human body. The human body  is comprised of interrelated /interconnected complexity  of multisystems. 

Author Response

Regarding the comment on Preiser's disease, we added text as suggested. 

The legends to the figures were connected to the diagnosis as suggested by the reviewer.